# Psychological Distress, Burnout, and Academic Performance in First Year College Students

**DOI:** 10.3390/ijerph19063356

**Published:** 2022-03-12

**Authors:** Jaume-Miquel March-Amengual, Irene Cambra Badii, Joan-Carles Casas-Baroy, Cristina Altarriba, Anna Comella Company, Ramon Pujol-Farriols, Josep-Eladi Baños, Paola Galbany-Estragués, Agustí Comella Cayuela

**Affiliations:** 1Chair in Medical Education, Universitat de Vic–Universitat Central de Catalunya, 08500 Vic, Spain; jaume.march@uvic.cat (J.-M.M.-A.); joancarles.casas@uvic.cat (J.-C.C.-B.); cristina.altarriba@uvic.cat (C.A.); anna.comella@uvic.cat (A.C.C.); ramon.pujol@umedicina.cat (R.P.-F.); agusti.comella@uvic.cat (A.C.C.); 2Group on Methodology, Methods, Models and Outcomes of Health and Social Sciences (M3O), Universitat de Vic–Universitat Central de Catalunya, 08500 Vic, Spain; paola.galbany@uvic.cat; 3Chair in Bioethics, Universitat de Vic–Universitat Central de Catalunya, 08500 Vic, Spain; 4School of Medicine, Universitat de Vic–Universitat Central de Catalunya, 08500 Vic, Spain; josepeladi.banos@uvic.cat

**Keywords:** burnout, psychological distress, academic performance, mental health, university students, medical students

## Abstract

Background: The first years of university can be very challenging for students. Previous research has focused on the study of the prevalence of burnout and of psychological distress in medical students. The aim of this study was to investigate the prevalence of psychological symptoms and burnout reported by first-year students, the relationship between these variables and their academic performance, and the differences between health and non-health sciences students. Methods: An observational study with a cross-sectional design was performed. Students of health sciences (medicine, nursing, physiotherapy, psychology), and non-health sciences (biology, social sciences, business management, and engineering) undergraduate programs completed the Brief Symptom Inventory (BSI-18) and the Maslach Burnout Inventory-Student Survey (MBI-SS). Students’ grades for the first semester were collected. Results: A sample of 506 students participated. Prevalence of psychological distress was 27.1% and burnout was 7.3%. Academic performance was unaffected in relation to either psychological distress or burnout. Non-health sciences students showed a greater risk of depression. Conclusions: This study provides evidence of the high prevalence of psychological distress in the first year of college. Even when burnout prevalence was low, the results suggest the need to introduce prevention programs to improve the psychological wellbeing of these students.

## 1. Introduction

The first year of university can be very challenging for students [1,2]. The change from secondary school to university, the academic pressure between jobs and exams, the lack of family support, and also economic factors can trigger a high level of stress in students that could lead to emotional exhaustion [3].

General psychological discomfort such as somatization (having trouble getting breath, feeling weak, having nauseas or chest pain without medical reasons), anxiety (nervousness, feeling fearful or having spells of panic), or depression (feeling no interest in things, hopeless about future or feeling of worthlessness), are common in university students [4,5,6], and they can also be predictors of poor academic performance [7].

Psychological distress often appears associated with burnout, and cognitive and behavioral problems [8]. Burnout is a multifaceted construct characterized by various degrees of emotional exhaustion (the feeling of not being able to give the best, both physically and psychologically), depersonalization (a negative or distance attitude towards other people, also defined as cynicism or disbelief), and a low sense of personal accomplishment (the tendency to feel incompetent) [9,10]. Academic burnout appears in students with a high emotional exhaustion and depersonalization together with low feelings of personal accomplishment related to their academic work [11,12].

Burnout does not only depend on individual predisposing factors, such as emotional expression or inadequate coping strategies for stress [13], but it is also associated with external factors such as the educational system [14]. It may have adverse effects by increasing distress and decreasing academic engagement [15], and it can be a good predictor of poor academic performance [16].

In recent years, several studies have shown a high prevalence of burnout among college students [11,17,18,19,20,21], particularly in medical students even in their first university years [22,23,24,25]. These findings could be justified by the high requirements to enter medical school and contact with patients [26,27,28].

In Spain, some studies have analyzed the prevalence of burnout both in undergraduate [3,11,13,14,29,30,31,32] and PhD students [33]. None of these studies has analyzed the relationship of burnout with psychological distress in university students. To the best of our knowledge, studies that compare psychological symptoms, academic burnout, and academic performance in university students are also lacking.

The aim of the present study was to determine the prevalence of psychological distress and burnout reported by students of the first year in a Spanish university, as well as its relationship with academic performance.

## 2. Materials and Methods

### 2.1. Study Population

This study surveyed two cohorts (2018–2019 and 2019–2020) of all first-year students of undergraduate programs of health sciences (medicine, nursing, physiotherapy, psychology) and non-health sciences (social education, biology, biotechnology, business management, journalism, automotive engineering and mechatronics engineering). The classification of disciplines was made according to the International Standard Classification of Occupations [34]. Inclusion criteria were enrollment for the first time at the university and consent to participate in the study. Exclusion criteria were having language difficulties in understanding the questionnaire.

### 2.2. Study Design

An observational study with a cross-sectional design was carried out. The variables which were considered were sociodemographic (age, gender, hometown population, secondary school, university funding, university access route, and type of undergraduate studies), academic burnout, psychological distress, and academic performance.

For the detection of psychological distress, the Brief Symptom Inventory (BSI-18) questionnaire was used. BSI-18 is a brief self-applied inventory of psychological symptoms that has been validated in Spanish [35,36]. It has 18 items on the three subscales, somatization (6 items), anxiety (6 items) and depression (6 items); each item describes a symptom for respondents to rate on a Likert scale (0–4) according to how much the symptom has bothered them in the past week. The value of the Global Severity Index (GSI) is a general indicator of clinical psychological distress. This condition is considered when the global T score or the T score of two of the three subscales is equal to or higher than 63 points.

To measure burnout, we used the Maslach Burnout Inventory-Student Survey (MBI-SS) questionnaire [11] in its Spanish version [37]. This instrument is an adaptation of the Maslach Burnout Inventory questionnaire validated for students, and has been shown to have adequate reliability and factor validity [37,38]. The MBI-SS consists of 15 questions that evaluate the three dimensions of burnout: emotional exhaustion (five items), depersonalization (four items), and personal accomplishment (six items). Reponses to each item were measured by a 6-point Likert scale from 1 (never) to 6 (every day). For the correction of the test, we used the technical note *NTP-732* of the National Institute of Safety and Hygiene at Work of Spain [39]. This instrument defines the levels of burnout according to the percentiles obtained in such a way that the first quartile groups represent the low values in each of the dimensions, the second and third quartiles the average values, and the fourth quartile the highest values. According to the scores obtained, the three dimensions were classified as follows: Emotional exhaustion: low: 0–1.2; moderate: 1.3–2.8; high: >2.9. Depersonalization: low: 0–0.5; moderate: 0.6–2.25; high: >2.26 and personal accomplishment: low: <3.83; moderate: 3.84–5.16; high: >5.17. Burnout was classified using the two-dimensional definition that includes high levels of emotional exhaustion and depersonalization [40,41]. Personal accomplishment was not included for the burnout measurement since it is the dimension least related to the other variables and because it would have a less relevant and more irregular role in the appearance of burnout [42].

For the academic performance variable, we assessed students’ grades for the first semester (February) obtained from academic records. Grades are the weighted average of the subjects taken according to the European credits (ECTS), and scored on a 10-point scale, with a pass score established at 5 points.

We pilot-tested the complete survey on 15 students in higher undergraduate years and 2 psychologists who were also members of the research team. Based on this feedback, we reformatted the survey to improve its clarity.

### 2.3. Data Collection

We used the *EncuestaFacil* platform (www.encuestafacil.com; accessed on 14 February 2022) to conduct the survey. We ensured that each respondent provided only one response by giving them a personalized access code while guaranteeing anonymity.

Before starting the survey, students read an information sheet explaining the background, aims, and procedure of the study. They were informed that the ethics committee of the university had approved the study protocol and that their participation was voluntary. Informed consent was requested, ensuring the anonymity and confidentiality of the information at all times. Likewise, they were optionally offered feedback with an interpretation of the results and a set of tips that they could consult with the same access code as for the questionnaire.

The local research ethics committee approved the study protocol (reference no. 60/2018). The study was performed according to the Declaration of Helsinki. Data confidentiality was ensured according to local laws on the protection of personal data.

### 2.4. Data Analysis

We used descriptive statistics to show participants’ demographic characteristics, the prevalence of academic burnout, psychological distress scores, and academic performance grades.

For quantitative variables, mean, SD, and range were used and the 95% confidence intervals are presented, provided that the variable has a normal distribution. The chi-square test was used to relate the nominal variables; Kolmogorov–Smirnov test was used for normality tests; Mann–Whitney U test was used for comparisons of two means with no normal distribution; Student–Fisher *t* test was used in groups; Kruskal–Wallis test was used for the comparison of more than two averages with no normal data. All analyzes were performed using the IBM SPSS version 28.0 for Windows (SPSS Inc., Chicago, IL, USA).

## 3. Results

A sample of 506 out of a population of 1481 students (overall response rate 34.2%, error sample 3.5%, NC 95%) completed the questionnaire online between October and November, both in 2018 and 2019. General demographic characteristics are shown in Table 1. In the first cohort there were 301 responses (response rate 42.4%) and in the second cohort there were 205 (response rate 26.6%). No statistical differences were found among sociodemographic variables between the two cohorts, as in the comparison of health sciences and non-health sciences students. Women accounted for 64.8% of the respondents; a higher proportion of women existed in all undergraduate programs except Automotive and Mechatronics engineering, and Business management. Mean age did not differ between groups of students of the different programs.

### 3.1. Prevalence of Psychological Distress

It was found that 27.1% (137 of 506) of the students meet the criteria for psychological distress (Table 2). More specifically, 28.7% (94) of the women and 24.2% (43) of the men meet these criteria, and no significant differences were found according to gender (Chi 1.184; *p* = 0.277). Health sciences students presented more clinical psychological distress than non-health sciences students (*p* = 0.002). No significant differences were observed between psychological distress and sociodemographic variables.

Regarding the three subscales of psychological distress, health sciences students reported more clinical somatization (*p* = 0.004) and clinical anxiety (*p* = 0.001) than non-health science students, whereas no differences were observed in relation to depression.

Regarding the gender distribution, 21.0% (69) of women and 18.0% (32) of men met the criteria of somatization subscale, 25.6% (84) of women and 29.2% (52) of men met the criteria of depression subscale, and 27.4% (90) of women and 23.0% (41) of men met the criteria of anxiety subscale. No significant differences were found according to gender in any of the subscales.

### 3.2. Prevalence of Academic Burnout

Among respondents, 7.3% (37) of students met the criteria for academic burnout (Table 3). Of these students, 5.8% (19) were women and 10,1% (18) were men. No significant differences were found according to gender (Chi 3.177; *p* = 0.075). No significant differences were found between burnout prevalence and sociodemographic variables, except for the relationship with study funding (Chi 5.467, *p* = 0.046). For 94.6% (35) of students with burnout, the family was their source of funding. This percentage was 79.2% (369) in students without burnout.

Regarding the two subscales of academic burnout, 45.3% (229) of the students showed emotional exhaustion and 8.5% (43) showed depersonalization. Regarding gender and the subscales, 45.7% (150) of women and 44.4% (79) of men presented emotional exhaustion, but these differences were not significant (Chi 0.085; *p* = 0.771). Additionally, 6.1% (20) of women and 12.9% (23) of men presented depersonalization. Significant differences were found between genders in the level of depersonalization (higher in men, Chi 6.909; *p* = 0.009).

No significant differences were found between health sciences students and non-health sciences students and academic burnout, in either of the two subscales.

### 3.3. Academic Performance

Data are missing for two health sciences students and six non-health sciences students due to withdrawal of studies, so in this sample *n* = 498. The average academic mark obtained at the end of the first semester of the first year was 6.79 (Table 4). Significant differences were only observed when academic performance average was compared with gender (women presented higher academic performance; mean mark scores 6.86 versus men 6.66; *p* = 0.005). Health sciences students showed higher academic performance than non-health sciences students (*p* < 0.001).

### 3.4. Psychological Distress, Burnout, and Academic Performance

When analyzing these three variables (*n* = 498), 3.4% (17) students reported psychological distress and academic burnout simultaneously; 23.3% (116) students presented only psychological distress and 3.6% (18) only academic burnout; while 69.7% (347) students did not meet the criteria for psychological distress or burnout.

Students with psychological distress had a higher prevalence of burnout (Chi square 9.194, *p* = 0.002). Of the students who presented psychological distress, 12.8% (17) presented burnout, compared to 4.9% (18) who did not have it and presented burnout.

Regarding the three subscales of psychological distress, we also have observed a higher prevalence of academic burnout: 2.4% (12) students had somatization and academic burnout (Chi square 5.2634, *p* = 0.022), 3.8% (19) had depression and academic burnout (Chi square 14.628, *p* < 0.001); 18 (3.6%) for anxiety and academic burnout (Chi square 13.047, *p* < 0.001).

Regarding the two subscales of academic burnout, of all the students with psychological distress (137), 69.3% (95) showed high levels of emotional exhaustion, 25.5% (35) medium levels, and 5.1% (7) lower levels. Additionally, of all students with psychological distress, 14.6% (20) showed high levels of depersonalization, 29.9% (41) medium levels, and 55.5% (76) lower levels.

Regarding academic performance, it was only lower in the group of women with clinical somatization criteria (*p* = 0.029). When comparing the mean grades of academic performance with the presence of psychological symptoms and burnout, no significant differences were found (Table 5). Academic performance did not show significant differences in terms of presenting or not presenting psychological distress or with regard to gender, or in the somatization or anxiety subscales.

In the binary logistic regression analysis, no significant correlations were observed between the sociodemographic variables and burnout and psychological distress as a predictive method. No associations have been found through a Pearson correlation between age, psychological distress (T global test punctuation), burnout, and academic performance (global test punctuation).

## 4. Discussion

Our study assessed the prevalence of psychological symptoms and burnout in first-year college students and found that this population may be at an increased risk of clinical depression and anxiety.

Previous studies have shown that the prevalence of depression can vary from 12.7% to 21.5% [43], or from 20% to 50% [5,44]. Our results for psychological distress (27.1%) are consistent with the prevalence of depression in previous studies from Spain (30.0%) [45] and Saudi Arabia (31%) [4]. These studies revealed worrisome rates of prevalence of depression, similar to previous research with other instruments, such as with the Patient Health Questionnaire-2: 28.5% in first and third-year medical students in United States [46], and in 30.3% of second, fourth, and fifth years of dentistry school in Spain [47]. Additionally, a systematic review gave a prevalence rate of 27.2% of depression in medical students [48].

No significant differences were found between psychological symptoms and gender. In a comparison of health sciences with non-health sciences students, we found significant differences in the prevalence of psychological symptoms which is in line with previous studies that suggest it is higher in health sciences studies [6,49]. Furthermore, health sciences students presented more clinical somatization and clinical anxiety.

Previous research has focused mainly on the prevalence of psychological distress on medical students [5,49]. However, this can be extended to students of all health sciences. High levels of emotional distress in our health sciences students may be due to the personal reaction to the stress of an intense study load and also to factors specific to the Medicine or Health Sciences curriculum [50,51].

Regarding the prevalence of burnout, it is striking that the percentages observed in our population (7.3%) are one of the lowest in relation with previous studies. This could be explained because the sample focused only on first-year students, and the assessment of burnout was performed using the two-dimensional definition and not the original three-dimensional one [40,41]. However, considering that the measurement was carried out during the first months of the first trimester in the university, this lower prevalence of burnout could mean that academic tasks are perceived as a challenge rather than a threat.

No significant differences were found between academic burnout and gender, except for the subscale of depersonalization. Although some research show empirical evidence that women suffer more burnout than men [9], other studies conclude the opposite [52,53]. However, it is agreed that women tend to show significantly higher levels of emotional exhaustion and lack of personal fulfillment at work, and men show higher levels of depersonalization [54]. This could be explained by gender stereotypes that consider women to be more emotionally sensitive [55].

Regarding our results, burnout is not significantly different between health sciences students and other undergraduates’ programs. The focus of previous investigations highlights the impact of academic burnout in first-year medical students [5,44] and extends the concern towards all the health sciences because academic burnout predominantly affects those professions involved in continuous and direct contact with people [56]. Previous research showed that there is a high prevalence of burnout in health sciences students, such as dental [17,47], nursing [57,58,59], and medical students [26,41,60,61,62]. The prevalence of burnout, for example in medical students, in these studies, can vary from 7% to 75.2% [63,64].

Previous research in Spain measured academic burnout in preclinical medical students and observed a range of prevalence from 14.8% [60] to 20.9% [41]. In other studies, such as one carried out in the United States, it was found that 11.5% of first-year medical students had high scores on all three dimensions of burnout [65], a percentage similar to our findings.

Students with psychological distress had a higher prevalence of burnout. It is important to mention that burnout did not predict psychological symptoms or academic performance, unlike previous research [3,15,66,67]. A very low percentage of students met the criteria for clinical burnout and psychological distress (3.4%). It is important to mention that all those who have burnout have psychological distress, but not all those who have psychological symptoms have academic burnout.

Regarding academic performance, we found that it has associations neither with psychological distress nor burnout, so it could not be an indicator of discomfort, unlike previous research has reported [15]. However, we have found that depression criteria affect academic performance in women more than men.

First-year college students had more psychological distress than burnout. As we have pointed out, academic burnout is relatively low in this sample, including health sciences students. However, as previous studies have established, a greater risk for burnout can be expected as medical students progress through medical school [68].

To confirm whether this psychological distress leads to burnout, or how psychological symptoms and burnout develop throughout all university courses, future longitudinal studies are required. For now, we assume that detecting students with psychological distress early on is one of the most important measures to improve student well-being, and also taking into account the importance of researching students on all university degrees, not just those in health sciences.

Our study has some limitations. The major limitation of burnout studies is the lack of a standard definition of the syndrome when using the MBI-SS. In the present study, the two-dimensional definition was used (high levels of emotional exhaustion and high levels of depersonalization) as in previous studies [40,41,42], even when it may result in an overestimation of burnout rates [41,69].

On the other hand, as with other self-reporting questionnaires, MBI-SS and BSI-18 may not detect some cases of burnout or psychological symptoms due to the fact that they are self-administered. In-depth interviews would be necessary in some cases, although the sample of students that we could reach with this method would be much smaller.

Moreover, the selection of the sample was made with volunteer students who reached one-third of the whole first-year student body. This may indicate that the students who answered the questionnaire had a higher degree of maturity or responsibility, which could influence the results of psychological distress or burnout.

In future research, it would be necessary to explore the possibility of evaluating twice a year, i.e., at the beginning and end of the academic year. Future research is needed also to predict psychological symptoms and burnout evolution over years of study, with special focus on non-health sciences students. Additionally, it would be interesting to be able to conduct broader, multicenter studies, in different Spanish communities and from other countries.

In future research, it could also be important to analyze the relationship between psychological distress, burnout, and academic performance with extracurricular activities [64], and other patterns of behavior such as alcoholism [70], eating habits, inadequate sleep or exercise [71], obsessive behaviors, and suicidal ideation [72]. Further studies considering the COVID-19 pandemic need to be analyzed [73,74,75,76,77] in order to compare the data obtained in pre-pandemic times with data collected during the pandemic and post-pandemic.

## 5. Conclusions

This study aimed to examine the prevalence of psychological distress and burnout and its association with burnout and academic performance among first-year university students. The results showed the existence of significant psychological discomfort in all students, not only those studying health sciences, as in previous research. Universities should not only monitor students through academic performance, but they should also concern themselves with the emotional state of their students, which must evolve at an optimal academic level while ensuring a better quality of life during their university education.

## Figures and Tables

**Table 1 ijerph-19-03356-t001:** Sociodemographic characteristics of the study sample (*n* = 506).

**Age, Mean (SD)**	19.2 (3.06)
**Gender, *n* (%)**	
Men	178 (35.2)
Women	328 (64.8)
**Hometown population, *n* (%)**	
Less than 10,000 people	124 (26.4)
10,000–50,000 people	202 (43.0)
More than 50,000 people	144 (30.6)
**Secondary school, *n* (%)**	
Public	308 (61.1)
Private	196 (38.9)
**University funding, *n* (%)**	
Family	404 (80.3)
Scholarship	26 (5.2)
Summer job	24 (4.8)
Year-round work	49 (9.7)
**University access routes, *n* (%)**	
University entrance examinations	326 (68.1)
Higher professional training	114 (23.8)
Other	39 (8.1)
**Program, *n* (%)**	
**Health sciences**	276 (54.5)
Medicine	78 (15.4)
Nursing	47 (9.3)
Physiotherapy	51 (10.1)
Psychology	100 (19.8)
**Non-health sciences**	230 (45.5)
Biology	53 (10.5)
Biotechnology	47 (9.3)
Business management	18 (3.6)
Social education	26 (5.1)
Journalism	18 (3.6)
Automotive engineering	42 (8.3)
Mechatronics engineering	26 (5.1)

**Table 2 ijerph-19-03356-t002:** Prevalence of psychological distress (*n* = 506). Values are expressed as number of students, and percentages are in brackets.

University Program *n* (%)	Psychological Distress			GSI
	Clinical Somatization	Clinical Depression	Clinical Anxiety	
**Health sciences ****	68 (24.6)	82 (29.7)	88 (31.9)	90 (32.6)
Medicine	15 (19.2)	23 (29.5)	23 (29.5)	25 (32.1)
Nursing	9 (19.1)	11 (23.4)	11 (23.4)	11 (23.4)
Physiotherapy	12 (23.5)	11 (21.6)	11 (21.6)	10 (19.6)
Psychology	32 (32.0)	37 (37.0)	43 (43.0)	44 (44.0)
**Non-health sciences ****	33 (14.3)	54 (23.5)	43 (18.7)	47 (20.4)
Biology	5 (9.4)	13 (24.5)	9 (17.0)	9 (17.0)
Biotechnology	6 (12.8)	10 (21.3)	7 (14.9)	9 (19.1)
Business management	2 (11.1)	3 (16.7)	3 (16.7)	3 (16.7)
Social education	5 (19.2)	2 (7.7)	2 (7.7)	4 (15.4)
Journalism	5 (27.8)	8 (44.4)	7 (38.9)	6 (33.3)
Automotive engineering	3 (7.1)	10 (23.8)	7 (16.7)	7 (16.7)
Mechatronics engineering	7 (26.9)	8 (30.8)	8 (30.8)	9 (34.6)
**** *p*-value**	0.004	0.115	0.001	0.002
**Overall**	101 (20.0)	136 (26.9)	131 (25.9)	137 (27.1)

** Comparison between health sciences and non-health sciences students. Chi-square test.

**Table 3 ijerph-19-03356-t003:** Prevalence of academic burnout (*n* = 506). Values are expressed as number of students, and percentages in brackets.

University Program *n* (%)	Academic Burnout Subscales		Academic Burnout
	Emotional Exhaustion (I)	Depersonalization (II)	
**Health sciences ****	130 (47.1)	20 (7.2)	17 (6.2)
Medicine	43 (55.1)	6 (7.7)	4 (5.1)
Nursing	27 (57.4)	2 (4.3)	2 (4.3)
Physiotherapy	19 (37.3)	3 (5.9)	3 (5.9)
Psychology	41 (41.0)	9 (9.0)	8 (8.0)
**Non-health sciences ****	99 (43.0)	23 (10.0)	20 (8.7)
Biology	26 (49.1)	7 (13.2)	4 (7.5)
Biotechnology	22 (46.8)	4 (8.5)	4 (8.5)
Business management	5 (27.8)	1 (5.6)	1 (5.6)
Social education	7 (26.9)	2 (7.7)	2 (7.7)
Journalism	13 (72.2)	0	0
Automotive engineering	16 (38.1)	4 (9.5)	4 (9.5)
Mechatronics engineering	10 (38.5)	5 (19.2)	5 (19.2)
**** *p*-value**	0.361	0.269	0.275
**Overall**	229 (45.3)	43 (8.5)	37 (7.3)

** Chi-square test.

**Table 4 ijerph-19-03356-t004:** Academic performance average (*n* = 498). Values are expressed as mean mark scores and number of students in brackets.

University Program	Mean Mark Scores (*n*)
**Health sciences ****	6.97 (274)
Medicine	6.88 (79)
Nursing	7.01 (46)
Physiotherapy	6.97 (51)
Psychology	7.04 (99)
**Non** **-health sciences ****	6.56 (224)
Biology	6.53 (52)
Biotechnology	6.79 (44)
Business management	7.08 (18)
Social education	6.82 (25)
Journalism	6.42 (18)
Automotive engineering	6.35 (42)
Mechatronics engineering	5.98 (24)
**** *p*-value**	<0.001
**Overall**	6.79 (498)

** Mann–Whitney U test

**Table 5 ijerph-19-03356-t005:** Psychological distress, burnout, and academic performance (*n* = 498). Values are expressed as mean marks with numbers of students in brackets.

Variables	Academic Performance Average (*n*)		*p*-Value ***
Without psychological distress nor burnout	6.76 (347) SD 0.97		
Psychological distress (only)	6.89 (116) SD 0.75		
Academic burnout (only)	6.59 (18) SD 1.80		
Psychological distress and burnout simultaneously	6.73 (17) SD 0.79		
*p*-value ****	0.454		
	**Health sciences**	**Non-health sciences**	
Without psychological distress nor burnout	6.94 (179) SD 0.83	6.58 (168) SD 1.08	<0.001
Psychological distress (only)	7.05 (78) SD 0.73	6.56 (38) SD 0.70	<0.001
Academic burnout (only)	6.96 (6) SD 0.43	6.39 (12) SD 2.19	0.820
Psychological distress and burnout simultaneously)	7.00 (11) SD 0.72	6.24 (6) SD 0.72	0.122
*p*-value **	0.486	0.299	

* Mann–Whitney U test. ** Kruskal–Wallis test.

## Data Availability

Authors agree to make data and materials supporting the results or analyses presented in their paper available upon reasonable request.

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
