# Peer review of "Psychological Distress, Burnout, and Academic Performance in First Year College Students"

_ijerph, 2022, doi:10.3390/ijerph19063356_

Round 1

Reviewer 1 Report

This is a workmanlike and well constructed paper. It is on the important topic of student mental health and will be of interest to researchers in the field and academics in general. The methodology is very thoroughly and clearly explained and the only English error I could spot was on P 3 line  102 evaluate (no s).

I have few suggestions for minor changes . Depersonalisation and somatisation  are not words that are widely known outside medical science and psychology . The authors should briefly define  depersonalisation  and indicate what symptoms were involved in the somatisation measure  It is interesting that the only statistically  significant gender difference was in depersonalisation. This heightens the need to explain it.

In the discussion section,  the authors state that women are more at risk, but that seems at odds with previous statements that there are no significant  gender variations. Can the authors make clear the basis on which they come to this conclusion? 

On P7 I note that there is a very  low withdrawal rate from the cohort . But this may relate to the self selection of the sample  which only constitutes one third of the whole first-year student body. It is very likely that the respondents were the more mature and serious students who are less liable to drop out. This may also explain why the percentage experiencing burnout was lower than in previous studies. I think the authors should acknowledge this as a limitation.

 The finding  that there is more psychological distress among the health science students does not surprise me. Medicine in particular  is a very challenging and demanding course of study. The requirements of lab work, exposure to  distressing material and clinical procedures  and the need for quantitative skills may  lead to higher anxiety and stress. If there is information on this in the other studies of health sciences students which are  referenced, it would be useful to include it briefly.

This points to what is the general limitation of this research; it tells  us the what but not the why. Some qualitative work to establish what causes depression and anxiety in these students would be a useful next step  - my own research suggests homesickness, worry about performance and the difficulty of adjusting to more independent methods of study than experienced at school are key factors. Also in the U.K. there is a strong party culture among first year students which may lead to difficulties with studying, and involves use of alcohol and drugs which might well contribute to depersonalisation.

Despite this, this an informative and well crafted piece and I would   recommend it for publication.

Reviewer 2 Report

The manuscript by March-Amengual et al. deals with a very important topic. Overall the manuscript is clearly written. However, a few things should be adressed:

  1. The formatting of the tables makes it difficult to comprehend
  2. Please provide link to EncuestaFacil platform
  3. Can you provide any reference to support the division of included disciplines into health and non-health sciences?
  4. lines 208-208 mention this "this criterion" . Please make it clear what criterion are you talking about.
  5. percents presented - please make it sure that you consistently report percents from the whole sample; right now I have the feeling that some percents refer to subsamples of certain groups, like in lines 212-219.
  6. Please rephrase unclear formulations of results like: "Significant differences were  found between gender and depersonalization" - this should be "between genders in the level of depersonalization"?
  7. why no relationships or predictions were planned by means of correlational or regression analyses?

Round 2

Reviewer 2 Report

Authors have addressed my comments.